# Understanding the One Belt One Road Initiative (BRI) influence on exportations of Chinese smartphones: The moderating role of the GDP per capita

Karamoko N'da[1]*, Jiaoju Ge[1], Steven Ji-Fan Ren[1], Jia Wang[2]

1 School of Economics and Management, Harbin Institute of Technology (Shenzhen), Shenzhen, China,
2 School of E-Commerce and Logistics, Suzhou Institute of Trade & Commerce, Suzhou, China

☯ These authors contributed equally to this work.
* andabuy@yahoo.com

## Abstract

The One Belt One Road Initiative (BRI) has been the subject of multitudinous studies from various angles. Most previous studies have focused on BRI's economic, geopolitical, or commercial implications for China. However, the few studies that focused on BRI's influence on the exportations or importations of Chinese products via the Chinese Cross-border Electronic Commerce Market (CCBECM) have been carried out based only on authors' opinions rather than on empirical evidence. Therefore, the actual effect of BRI on the exportations of Chinese product brands via CCBECM in BRI countries still needs to be discovered. Utilizing B2C exportation data of Chinese smartphones and a Difference-in Difference Model (DIDM), we have first examined the actual and direct impact of BRI policy on Chinese smartphone brands exportations via the Chinese Cross-border Electronic Commerce Market (CCBECM) from 2012 to 2019 in BRI countries. Secondly, we assessed the moderating role of GDP per capita (GDP) and Internet Access Rate (IAR) between BRI policy and exportations of Chinese smartphone brands. The results showed that the impact of BRI remains insignificant on the exportations of Chinese smartphones via CCBECM in BRI countries. However, it could be significant if BRI includes more developed and economically strong countries. The study also highlighted a negative moderating role of GDP per capita between BRI policy and exportations, showing that the higher the BRI effect is, the less GDP per capita will influence Chinese smartphone exportations in BRI countries.

## Introduction

With the lack of local industries and suppliers able to satisfy local demands, most consumers in emerging and developing countries rely only on Cross-Border E-Commerce (CBEC) markets to purchase lacking products on local markets. As a result, CBEC has become an essential transaction means between buyers and sellers worldwide. It is also an essential factor of economic growth for countries like China that understand the need to implement policies, such

**Funding:** This study is financially supported by the National Natural Science Foundation of China (Project No. 71402039 and Project No. 71831005), Shenzhen Peacock Program (Project No. KOCX2015032715503970). The funders had no role in study design, data collection and analysis, decision to publish, or preparation of the manuscript.

**Competing interests:** The authors have declared that no competing interests exist.

as the One Belt and Road Initiative (BRI), to boost that sector [1]. According to several scholars, the implementation of the BRI has provided a tremendous means for the growth of the Chinese Cross-border E-Commerce Market (CCBECM) [2] and excellent facilities for BRI countries in terms of easy access to Chinese products [3,4].

Initially announced by President Xi Jinping as a financial and trade cooperation project between China and countries members [3–5], BRI policy has been studied by many researchers from various angles, such as the geopolitics angle [6–11], economic and financial angle [12–17], infrastructural angle [18,19], cultural angle [20–22], energetic angle [23,24], environmental angle [25,26], and domestic politics angle [27]. However, some of those studies have strongly criticized the BRI policy, as they claim to see hidden diplomatic-geopolitical ambitions behind the project [7,8]. Those studies claimed that the BRI policy stake is to assert a more active foreign policy based on the new vision of the Chinese world order, which is interconnected capitalism. Therefore, BRI could be seen as an effort to strengthen China's diplomatic and political presence in BRI countries [28]. This point of view is also that of several world organizations led by the West, such as the G7, which describes the BRI as one of the essential links in "debt trap diplomacy" [29]. Therefore, on June 26th, 2022, during the penultimate G7 summit, the group of seven most industrialized countries pledged to raise about US $600 billion in funds over the next five years to counter the BRI initiative and replace it with the newly global project named "Partnership for Global Infrastructure and Investment" to finance needed infrastructure in developing countries [29].

However, other studies pointed out that the BRI primarily aims to support China's domestic economic growth by promoting sectors such as CCBECM to find new outlets and opportunities for Chinese industry and product brands [2–5]. Those studies claimed that implementing the BRI has contributed to boosting CCBECM. In this perspective, Hou et al. [2] stated that the advent of BRI has provided great opportunities for the growth of CCBECM. Mou et al. [30], argued that given the objective of BRI, which is to assist Chinese companies located along the BRI road to expand their market, the contribution of CCBECM will be necessary to achieve such an objective. Therefore, BRI could contribute to maintaining the increasing development of the CCBECM [31]. However, Ding et al. [32] consider that the contribution of BRI to the development of the CCBECM will depend on BRI countries' socioeconomic development through factors such as logistics infrastructures and internet access.

Theoretically, some other studies have tried to explain the influence of BRI on Chinese cross-border activities. Thus, in examining the connection between Beijing's investments in BRI nations and BRI nations' exportations of Chinese products, Ding et al [32] found that China's investments in BRI countries could theoretically explain Chinese product brand exportations through CCBECM. Thanks to investments in logistics and socio-economic infrastructures, the GDP of those countries increased, and those countries exported more Chinese products [32]. Therefore, BRI could bring significant opportunities to both China and BRI countries [5].

Yin and Choi [33] and Qi et al.[34] also argued that since implementing the BRI policy, CCBECM has experienced extremely rapid growth of about 27.03% of the growth rate every year. With that growth, China is now the first B2C CBEC market worldwide, with about 40% of the global share [35], especially in B2C sales of Chinese smartphones through the CBEC market [2]. However, despite the market's unprecedented growth since BRI policy implementation, literature has remained silent regarding the actual impact of BRI policy on the exportations of Chinese smartphone brands through B2C CCBECM in BRI countries. Therefore, this study intends to examine the actual effect of BRI policy on Chinese smartphone brands' exportations across BRI countries by investigating the following questions: Did the BRI policy influence the exportations of Chinese smartphone brands through B2C CCBECM in BRI countries

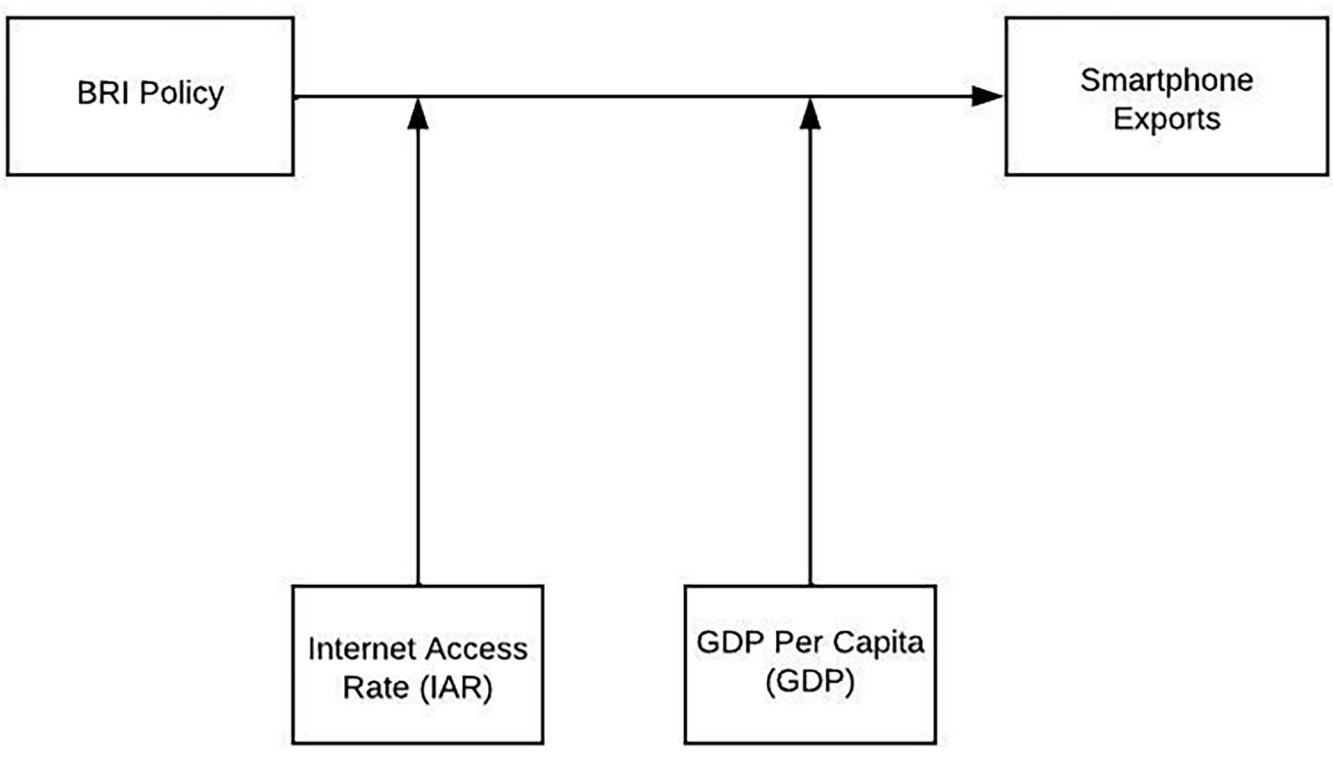

**Fig 1. Theoretical model.**

since the implementation of the BRI? If not, why is that so? What factors could moderate the BRI policy impact?

To provide answers to these research questions, we investigated, in addition to the BRI policy impact, the moderating effects of two macroeconomic factors (i.e., the Internet Access Rate (IAR) and GDP per capita (GDP)) between BRI policy and exportations of Chinese smartphone brands in BRI countries. In doing so, we avoid any diplomatic and geopolitical vision of the BRI policy and look only at its actual impact on the exportation of Chinese smartphones. Therefore, this study could provide valuable information to better understand the BRI policy impact on Chinese smartphone exportations through CCBECM in BRI countries. According to Hou et al. [2], among several Chinese product brand categories, Chinese smartphones are the most purchased through CCBECM after implementing BRI. Therefore, this current study focuses on Chinese smartphones exported through B2C CCBECM. The BRI project classifies countries into two groups: BRI countries and non-BRI countries. These two groups of countries are treatment groups and control groups, respectively [36]. Thus, utilizing a Difference-in Difference Model (DIDM), the result showed that the impact of the BRI policy on the exportations of Chinese smartphone brands via CBECM remains weak across BRI countries. However, it could be significant if the BRI could include more developed and economically strong countries. Fig 1 presents the theoretical model.

The remaining study is organized as follows: First, we examined the relevant literature related to BRI and CCBECM and analyzed how our study extends previous studies. Next, we presented the data description and developed a theoretical model, i.e., DIDM, to evaluate the impact of the BRI policy on exports of Chinese smartphone brands. Then, we analyzed and discussed the results. We ended the paper with theoretical contributions, conclusions and research limitations.

## 2. Literature review and hypotheses development

In recent years, with the increasing development of international trade between countries associated with the rapid digitalization of societies, consumers are increasingly shopping abroad [31]. As a result, the development of CCBECM has accelerated considerably [37]. Meanwhile, BRI, which is seen as China's new strategic cooperation policy intended to support China's cross-border activities, got much attention [38]. Most studies have analyzed the BRI project mainly from the perspectives of diplomatic, economic, and geopolitical deployment strategies [20–39]. According to those studies, the sole objective of this project is to promote China's diplomatic, economic, and geopolitical vision. Other studies also analyzed BRI's impact on Chinese international trade and found a positive effect, e.g. [38–41]. Moreover, in the literature, many authors' claims exist on the potential influence of the BRI policy on CCBECM. Some of these claims have indicated that BRI could reshape the dynamic of the CCBECM because the BRI has a strong influence on that market [31]. However, only some studies attempted to empirically verify the relationship between BRI policy impact and trade through CCBECM. The few studies that investigated that relationship focused on CCBECM and its related factors rather than BRI policy impact, e.i [30–32].

The literature on the relationship between BRI policy and CBECM could be classified mainly according to two perspectives: (1) authors' opinions, e.i, [2–32], and (2) empirical studies, e.i, [33]. Regarding authors' opinions, Hou et al. [2] and Xiao et al. [31] argued that the BRI had boosted Chinese international trade through CCBECM. Mou et al [30], examining buyers' repurchase intentions through CCBECM, argued that BRI could significantly support the CCBECM in assisting Chinese companies located along the Silk Road to enter the global market. However, all these previous studies failed to provide any empirical shreds of evidence to support their claims. Accordingly, previous authors' claims were more opinions than conclusions from empirical results. Therefore, the literature lacks empirical evidence regarding the BRI policy's actual impact on exports of Chinese product brands via CCBECM. That lack of empirical evidence may be due to the lack of data or past studies' incapability to propose a framework to empirically evaluate the actual impact of BRI policy on the export of Chinese products via the CCBECM.

To the best of our knowledge, the only study that investigated the relationship between BRI policy and CCBECM was the study of Yin and Choi [33]. After BRI implementation, they analyzed the impacts of China's cross-border online commerce on Chinese services and product exportations. They featured that CBEC decidedly affects service exportations more than products. However, it is critical to point out that the study of Yin and Choi [33] was not a study of the direct impact of BRI policy on CBEC exportations but rather an attempt to assess the export volumes of services and goods in BRI countries after BRI implementation.

To summarize: (1) although scholars have extensively argued that BRI policy has or could significantly impact Chinese international trade through CCBECM, authors have not provided sufficient and robust empirical evidence to support their claim. (2) A very limited number of studies investigated the direct effect of BRI policy on the exportations of Chinese smartphone brands via CCBECM in BRI countries. We believe there is a need for a deeper study to clarify the debate on the actual and direct impact of the BRI policy on the export of Chinese products, such as smartphones, via CBECM within BRI countries. Therefore, this study intends to examine the actual effect of the BRI policy on the exportations of Chinese smartphone brands in BRI countries through the CCBECM since the implementation of this policy.

### 2.1 BRI and Chinese smartphone exportations via CCBECM

As an international commercial trade between companies or individuals from various nations using international electronic marketplaces to purchase and sell while relying on the global

logistics network to deliver goods, CBEC results in significant cost savings on transaction costs [33]. Its substantial savings on transaction costs are most often related to communications, market research, and administration. Encouraged in emerging economies like China, CBEC has since enabled Chinese exporters and brands to overcome the constraints linked to isolation from potential markets, limited access to information, and high entry costs in the traditional international market [33]. With this, the CCBECM generated about 11 trillion RMB from its advent to 2019, thus creating thousands of dozens of jobs in China and outside China (as most of the sellers on CCBECMs, like Alibaba, AliExpress, Etc., are sellers from outside China). As a result, CCBECM has become a vital transaction means worldwide. Its impact on international trade was even more significant during the COVID-19 outbreak, when CCBECM had become almost the only international transaction framework in China, allowing suppliers to meet buyers' needs worldwide. It is, therefore, an industry essential to global and Chinese economic growth.

As a new Chinese cooperation model between countries members and China, BRI is designed to support reciprocal commercial preferentiality [36]. As a result, supporting CCBECM in BRI countries has been identified as one of the key goals of BRI policy [33]. That objective promotes CCBECM as part of the BRI implementation across BRI countries by constructing infrastructures such as warehouses [31–33]. Those infrastructures are set up to assist and provide more facilities for Chinese companies to export to BRI countries [33]. Since then, the Chinese government has been trying to gain new trade partners to maintain the development of its international trade [31–41]. Thus, one of BRI's missions from the beginning was to support the development of the CCBECM, which is known to affect the growth of Chinese international trade [32]. Therefore, studying BRI's impact on exports of Chinese smartphones through CCBECM can be considered an important research topic for BRI policy, Chinese companies and Chinese smartphone brands that strive to find new market opportunities in BRI countries [33–30].

Several researchers have seen the BRI as an essential policy for China at the economic and trade level, e.g. [31–41]. Since then, BRI policy has become a proliferating research subject in Chinese international offline and online trade literature [31]. Researchers stated that the contribution of CCBECM in the exports of Chinese product brands would be necessary to achieve the BRI objectives [33]. Those objectives consist of assisting Chinese companies and brands in expanding and finding new markets across BRI countries [29]. Yin and Choi [33] argued that since the BRI policy was put forward, the CCBECM has experienced extremely rapid growth by exporting Chinese manufactured products to BRI countries. Therefore, BRI could contribute to maintaining the growing exportation of Chinese products through the CCBECM [30].

In the case of offline international trade, most existing research found a positive effect of the BRI policy on Chinese product export activities. For instance, in simulating the impact of BRI policy on China's trade performance, [41] highlighted a positive effect of BRI policy on Chinese international trade promotion. However, they pointed out that without the impact of the BRI policy, China's exports would have remained constant. Yu et al. [40], utilizing a DIDM, highlighted that since BRI implementation, the index of bilateral trade preferences between China and BRI countries had increased about 8% faster than that of non-BRI countries. Exploring the potential impact of BRI policy on trade flows between China and ASEAN nations from 2000 to 2016, Foo et al. [38] has also found a positive and significant influence of the BRI policy on commercial flows.

This current study focuses exclusively on the B2C model of CCBECM. China is at the top B2C CBECM worldwide, with a market share estimated at $1,156bn in 2018 [35–42]. As a result, the B2C CCBECM has become a core pillar of China's international electronic trade [29–33]. That trade mode is considered the fastest way to export Chinese smartphones outside

China [43]. Allowing international consumers to access several types of Chinese product brands (e.g., electronic devices, smartphones, clothes, shoes, household goods, Etc.) [35]. Among those products, research showed that smartphones were the most exported through CCBECM [2–44]. However, despite those unprecedented exports of smartphones through CCBECM, little research has been interested in it. Therefore, we focused on Chinese smartphone products to carry out our investigation.

In the literature on the relationship between BRI policy and smartphone exports to BRI countries through CBECM, one notes several claims from previous studies without any empirical evidence. For instance, Yang and Asbury [45] claimed the advent of BRI provided great opportunities to Chinese smartphones regarding their purchases and promotions within-country members. In studying different countries' preferences for Chinese smartphone exportations via CCBECM, [2] argued that the advent of BRI has provided great opportunities for purchasing Chinese smartphone brands through CCBECM. However, despite all these claims, we note in the previous literature a lack of empirical studies demonstrating the impact of the BRI policy on Chinese smartphone exportations through the CCBECM in BRI countries.

Most often, the relationship between BRI policy and CCBECM activities associated with international trade has been explained from the perspective of DIDM (e.g. [36]). This model is one of the most popular policy evaluation methods and can easily isolate the "policy effect" from the "time effect" by eliminating errors caused by imperceptible time factors [36]. Based on that method, various studies on the impact of BRI policy on Chinese cross-border activities and international trade have reported a positive effect [40]. Yu et al. [40] highlighted a positive impact of the BRI policy on China's trade level in BRI countries. In examining BRI countries' participation in global value chains (GVCs), Wu et al. [36] found a positive and significant impact between BRI policy and BRI countries' participation in GVCs. Therefore, based on previous empirical research results, the BRI policy could positively influence the B2C export of Chinese smartphone products in BRI countries via CCBECM. Hence, we hypothesize that:

Hypotheses 1: BRI policy's implementation has positively influenced China's B2C exportations of smartphones.

## 2.2 Moderator effect: Internet Access Rate (IRA)

With the advent of the Internet and digital technologies, cross-border e-commerce is booming, allowing suppliers, sellers, and consumers to transact from different locations worldwide [46]. Research points out that countries intending to develop the CBEC industry need good internet coverage as a prerequisite [46,47]. That is because easier internet access within a country can encourage citizens to engage more in CBEC [46]. In this regard, Yin and Choi [33] defines internet access in the case of cross-border e-shopping as a means for international buyers to access e-commerce platforms to order products and have them delivered using the global logistics system. Internet access, therefore, can be seen as a prerequisite for the practice of the CBEC industry within a country.

The effect of the Internet on the e-commerce industry has been extensively examined in many studies. Furthermore, most have found positive relationships between the Internet and offline and online trade practices [47]. Nuruzzaman and Weber [46] showed that companies with internet access exported more than those lacking internet access. The report explains that exportations are increasing because the Internet makes it simpler for companies to interact with purchasers. In studying the Internet impact on bilateral commerce flows between 21 underdeveloped countries and OECD countries, Xing [48] shows that Internet access is crucial for developing CBEC in underdeveloped countries. Osnago and Tan, in the report of the World Trade Organization [47], highlighted that the Internet has a significant impact on

exportations. Based on bilateral trade data, Rodríguez-Crespo and Martínez-Zarzoso in the report of Nuruzzaman and Weber [46], found that internet use promotes product exportations. In exploring China's exportations through CCBECM to nations along the Belt Road, Ding et al. [32] showed internet access was among the top indicators influencing the exportation scales of BRI nations. Based on the literature reviewed here, it can be concluded that the Internet Access Rate (IRA) could be one of the key factors influencing cross-border exportations, whether online or offline.

In the BRI policy case, it is evident that most BRI countries are developing and emerging countries. That means their ability to access the Internet is a little tricky. Therefore, the ability of BRI countries to conduct cross-border electronic transactions may be less intense than that of countries with a high capacity to access the Internet. Hence, IAR may weaken international buyers' exportability of BRI countries due to the low level of their country's IAR. As a result, IAR may negatively alter the relationship between the BRI policy effect and Chinese product brand exportations through CCBECM. Therefore, we formulated the following hypothesis:

Hypothesize 2: The Internet Access Rate (IRA) moderates the influence of BRI policy on B2C exportations of Chinese smartphones via CCBECM in BRI countries, such that the relationship will be weakened when the IAR is lower.

## 2.3 Moderator effect: GDP per capita

Macroeconomic factors, such as GDP per capita, have been considered by several studies to be key factors in understanding and evaluating countries' exporting capacities in the cross-border e-commerce framework [49]. For that reason, the researchers concluded that CBEC transactions should be analyzed from the perspective of variables, such as the DGP per capita [49,50]. These researchers start from the reason that richness levels estimated by GDP per capita are fundamental for internet shopping practices in a country [51]. According to Lu [52], buyers always purchase according to their purchasing power. Purchasing power is measured at each country level from macroeconomic indicators, such as GDP per capita. Several academic research mentioned that GDP per capita influences e-commerce adoption within a country as it determines consumer purchasing power [53,54]. Anvari and Norouzi [53] and He and Wang [54] showed a positive influence of GDP per capita on CBEC transactions. Gibbs et al. [51] supported this view, showing that GDP per capita explains more than half of e-commerce sales variance. In studying exportations through CCBECM to nations along the Belt Road, Ding et al. [32] showed that GDP was among the top indicators influencing the exportation scales via CCBECM. Likewise, other studies showed that GDP per capita is an essential indicator for measuring the purchasing capacity of a product such as a smartphone worldwide [55,56]. Based on the above studies' findings, we believe GDP per capita can favour exportations of Chinese smartphone brands via CCBECM. However, as most BRI countries are developing and emerging countries, their GDP per capita could have a negative effect on their exportation capability. That implies that the capacity of consumers from BRI countries to conduct cross-border electronic transactions may be low due to their low level of wealth measured by the GDP per capita. Therefore, we believe that the BRI countries' GDP per capita level may negatively alter the relationship between the effect of the BRI policy and the B2C exportations of Chinese smartphones through the CCEBM. Therefore, we formulate the following hypothesis:

Hypothesize 3: The GDP per Capita moderates the influence of BRI policy on B2C exportations of Chinese smartphones via CCBECM in BRI countries, such that the relationship will be weakened when GDP per capita is low.

## 3. Method and material

### 3.1 Research methodology and theoretical model

To quantitatively ascertain the effect of BRI policy on the exportations of Chinese smartphones in BRI nations via the CCBECM, we used the DIDM. First revealed on September 7, 2013, by President Xi Jinping, the BRI project divided countries into BRI and non-BRI countries. Therefore, these two groups of countries can be seen as treatment and control groups, respectively [40]. Accordingly, the period before 2014 may be considered the pre-policy period, while the period from 2014 to 2019 is viewed as the post-policy period. To reach our goal, we first ran a standard OLS model. We first pooled the data before and after the policy implementation by ignoring the time variable. We did so to get the reference point for comparison to evaluate the BRI policy's effect. As highlighted by several scholars, macroeconomic indicators need to be taken into account when it comes to investigating countries' engagement in CBEC transactions [49]. We integrated countries' development level indicators into the equation to control differences between countries' economic situations. Hence, we specified the standard OLS regression equation as follows:

$$EXPORT_{it} = \alpha + \beta POLICY_{it} + \zeta x_{it} + \epsilon_{it} \tag{1}$$

Where $EXPORT_{it}$ is the Chinese smartphone exportations in OBORI countries via the CCBECM in year $t$. $POLICY_{it}$ is a binary variable that equals 1 if a sample country ($i$) is a member of BRI policy, and 0 otherwise. $\alpha$ is the constant term. $\beta$ represents the coefficient of the treatment group (BRI policy). $\zeta$ and $x_{it}$ are the coefficient and the variable of countries development levels (status). Afterwards, we ran the DID regression model by integrating the time variable. We described the DID regression as follows:

$$EXPORT_{it} = \alpha + \beta POLICY_{it} + \eta Postp_{it} + \theta(POLICY_{it} * Postp_{it}) + \gamma_n x_{it} + \epsilon_{it} \tag{2}$$

In this equation, $Postp_{it}$ is a binary variable that equal to 1 if $t$ is a post-policy period, and 0 otherwise; $POLICY_{it}*Postp_{it}$ is the interaction term, and equal to 1 if a country $i$ is a member of BRI policy (treatment group) during the post-policy period, and 0 if not. $\eta$ is the coefficient of the year fixed effects. $\gamma_n$ and $x_{it}$ are the coefficients and variables of *countries development level (status)* and $\epsilon_{it}$ represents the error terms.

The DIDM is based on two key assumptions: (1) the random sampling of the treatment group and (2) the parallel trend assumption. Concerning the first assumption, we believe that that assumption is satisfied. The BRI, a project intended to open China to the world, has integrated countries with diverse potentiality in politico-socio-economic power. For instance, as an emerging country and the second world military superpower, Russia is a member of the initiative, like small countries such as Bangladesh or Djibouti, to name a few. Therefore, there is no proof that China might have picked BRI nations dependent on their political or financial powers [40].

Regarding the second assumption (2), Fig 2 provides visual support, showing that the exportations of BRI and non-BRI countries (NBRI) have experienced almost the same exportation trends before and after implementing the BRI. One may observe that after the implementation of BRI (2014–2019), the exportations of the treatment group (BRI countries) and control group (non-BRI countries) have followed almost the same trends. However, exportation in non-BRI countries (control group) was higher than that of the treatment group (BRI countries).

After running the DIDM, we further ascertain the possible source of the differential influence of BRI policy on exports based on countries' socio-economic indicators. To this end, we

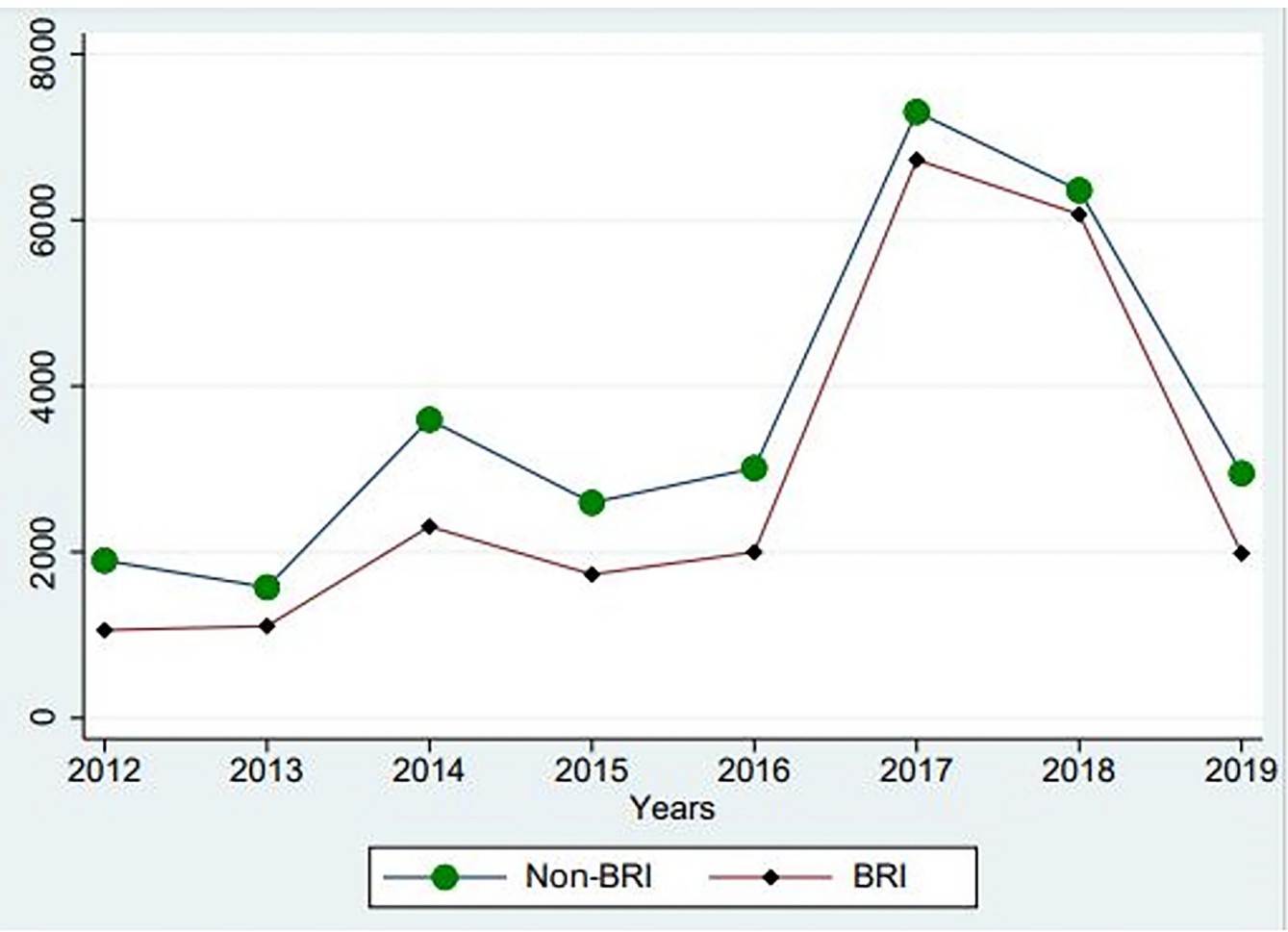

**Fig 2. BRI and non-BRI countries export trends.**

evaluated the moderating effect of two main socio-economic variables on exportations. Namely, IAR and their GDP per capita. As stated by Nolan et al. [49] and Li [57], these two variables constitute important pillars to understanding countries' exportations through a CBEC market.

## 3.2 Research context, data description and variable measurements

**Research context.** With BRI implementation, China has become one of the essential electronic markets, offering excellent opportunities to Chinese smartphone brands worldwide [2]. According to Hou el al. [2], Chinese smartphone brands are the most purchased worldwide among Chinese product brand categories in the context of BRI.

The market for Chinese smartphones is large. Out of the 76 brands of smartphones worldwide, China alone accounts for more than 12 smartphone brands, such as Huawei, Xiaomi, Meizu, Oppo, VIVO, and Coolpad [58]. In this current study, we focused on only three of them, i.e., Huawei, Xiaomi, and Oppo, because of their popularity within the CCBECM. These brands enter the global market with a strategy of affordable prices, which increases their market shares, especially across developing and emerging countries. For instance, in 2016, Huawei smartphones were sold in around 170 countries [59]. In the second quarter of 2018, Huawei

was ranked as the No.1 smartphone brand in China, with a 28.1% market share. By the end of 2018, Huawei and Xiaomi were among the best five-selling smartphone brands in the Chinese market [60]. At the same time, Huawei sold around 30.72 million smartphones with a 28.6% share, followed by Xiaomi, which sold about 12.61 million products with an 11.7% share. According to a report of Gartner [60], due to the global pandemic, while the worldwide smartphone market has experienced a decline in turnover of about 20%, Xiaomi has been the only brand that experienced growth in the first quarter of 2020 [60,61]. The report highlights that Xiaomi is the only smartphone brand that has managed to avoid a drop in turnover among the five major worldwide smartphone brands. Xiaomi recorded growth of 1.4% from 2019 to the first quarter of 2020, positioning itself as the third-largest phone brand globally ahead of Apple, with 12.1% of sales in the third quarter of 2020. Meanwhile, Huawei and Oppo sales decreased by 21.3% and 2.3%, respectively.

### 3.3 Data description and variables

Based on data at our disposal and BRI countries' list from the study of [14], 28 countries were set as the treatment or experimental group (BRI countries) and 41 non-BRI countries as the control group. The dataset consists of B2C export data via CCBECM from 2012 to 2019. The data used mainly comes from www.aliexpress.com and [62]. Through Aliexpress.com, Chinese product brands are sold online in B2C mode to international markets. The data was collected by Google's programming language and Webharvy's software. Data consists of 3 leading Chinese smartphone brands purchased by consumers from 69 countries (BRI and Non-BRI countries): Huawei, Xiaomi, and Oppo. Each brand is made up of several categories of mobile phones.

**Data source.** The data source for this study is the AliExpress platform, which we got from www.aliexpress.com. AliExpress is one of the platforms intended to promote CCBECM outside country borders. As an internationally leading platform for B2C sales, the AliExpress platform sells to about 150 million consumers from 190 countries worldwide [57]. With more than 2.4 billion visitors every year and over 100 million products [57]. It is made of about 1.1 million e-sellers located in China and outside China (Affiliated stores) [57]. Most of the sellers on the platform are retailers rather than manufacturers, and they obtain products from factories across China for reselling to international buyers [57]. Therefore, this platform somehow constitutes a gateway between Chinese smartphone brands and global consumers.

**The dependent variable.** This study examines whether the BRI policy has influenced the B2C exportations of Chinese smartphones through CCBECM in BRI countries since implementing the BRI policy. Thus, this article aims to look at the real impact of BRI policy on the exportation of Chinese smartphones into BRI nations. Furthermore, it provides empirical evidence to support or refute the opinions or claims of previous authors regarding the BRI's potential impact on exportations of Chinese smartphone brands in BRI countries. Therefore, the main dependent variable is exportations ("EXPORT"). The dependent variable measures the exportations towards BRI countries of the three leading Chinese smartphone product brands (e.i., Huawei, Xiaomi, and Oppo) through CCBECM from 2012 to 2019.

**The independent variable.** The main independent variable of the study is BRI policy ("Policy"). To measure the impact of the BRI policy, we used the list of BRI countries from the study of [14]. We did so because the Chinese government has not officially defined the definitive list of BRI countries until now [14]. The explanatory Variable (Policy) measures whether a country is a member of the BRI policy or not. Among countries covered by the data set, 28 BRI countries represent the treatment group (coded = 1), and 41 non-BRI nations represent the control group (coded = 0). The years (2012–2019) considered in this dataset include pre- and

**Table 1. Description of variables.**

| variables | Description | Sources |
|---|---|---|
| $EXPORT_{it}$ | The EXPORT level of Chinese smartphone brands by BRI countries via CCBECM | www.aliexpress.com |
| $\beta POLICY_{it}$ | BRI policy binary variable that controls for unobserved differences between groups | www.aliexpress.com |
| $\eta Postp_{it}$ | The time binary variable that controls for unobserved changes affecting both groups | www.aliexpress.com |
| $\theta(POLICY_{it}*Postp_{it})$ | The interaction term quantifying the effect of the treatment on the average outcome for purchase | www.aliexpress.com |
| $\gamma x_{it1}+\delta y_{it1}+\zeta z_{it1}$ | The set of Control variables controlling for observable differences within and between groups. | www.aliexpress.com |

post-BRI periods. The year 2014 is regarded as the starting year of the BRI. We coded the time: time = 0 if the year is a pre-BRI period; time = 1 if the year is a post-policy period. Another explanatory variable is the interaction term between the groups and time.

**Control variables.** The study also incorporated country development level ("Status"), Internet Access Rate (IAR), and GDP per capita (GDP) as the control variables. The World Bank classification of countries' development levels measured the country's development level. Thus, Lower Middle-Income countries and lower-income countries are coded as Developing countries and equal to 1; Middle-Income countries are coded as Emerging countries and equal to 2; and higher-income countries are coded as Developed countries and equal to 3. IAR and GDP per capita were obtained from the database of the [62]. Table 1 describes the study's main variables, and Table 2 details the summary statistics of the dataset. Moreover, Tables 3 and 4 show the results of DIDM and the analysis of the moderating variables, respectively.

# 4. Empirical results: OLS regression and Difference-in-Differences Model (DIDM)

To ascertain the effect of BRI policy, we first run a simple OLS regression described in Eq (1). And the results are reported in Table 3 (Model 1). The results of Model 1 in Table 3 shows that the effect of the BRI policy on export is not significant ($\beta$ = 17.5278, t. = 0.71, p>.05). However, concerning the two last control variables $x_{it2}$ and $x_{it3}$ representing respectively, the effect of emerging countries and developed countries they have a significant and positive impact on

**Table 2. Summary statistics.**

| Name of brands | Number of product brand purchases | | | | |
|---|---|---|---|---|---|
| **Huawei** | 17767 | | | | |
| **Oppo** | 17485 | | | | |
| **Xiaomi** | 17015 | | | | |
| **Variable** | Obs | Mean | Std. Dev. | Min | Max |
| $EXPORT_{it}$ | 621 | 93.383 | 252.177 | 4 | 3262 |
| $\beta POLICY_{it}$ | 621 | 0.406 | 0.491 | 0 | 1 |
| $\theta(POLICY_{it}*Postp_{it})$ | 621 | 0.309 | 0.463 | 0 | 1 |
| $\eta Postp_{it}$ | 621 | 0.771 | 0.421 | 0 | 1 |
| $\gamma x_{it}$ | 621 | 2.005 | 0.87 | 1 | 3 |
| $\delta y_{it}$ | 621 | 63.57 | 24.762 | 4.94 | 99.15 |
| $\zeta z_{it}$ | 621 | 21379.07 | 23548.2 | 802.518 | 118823.6 |
| **Country** | 621 | 35 | 19.932 | 1 | 69 |
| **Region** | 621 | - | - | 1 | 14 |
| **Years** | 621 | - | - | 2012 | 2020 |

**Table 3. Differences in differences model.**

|  | Model 1 |  |  | Model 2 |  |  |
|---|---|---|---|---|---|---|
| Variable | Coef. | T value | [95%Conf.Interval] | Coef. | T value | [95% Conf. Interval] |
| $POLICY_{it}$ | 17.528 (24.6396) | 0.71 | -30.8717 65.9274 | 1.049 (37.0001) | 0.03 | -71.6309 73.7293 |
| $x_{it1}$ | 36.133 (21.8142) | 1.66 | -6.7172 78.9823 | 2.129 (28.4676) | 0.07 | -53.7913 58.0475 |
| $x_{it2}$ | 135.867 (27.330)*** | 4.97 | 82.1822 189.5523 | 101.368 (32.7989)** | 3.09 | 36.9406 165.7959 |
| $x_{it3}$ | 107.575 (18.2715)*** | 5.89 | 71.6839 143.4654 | 77.052 (25.7455)** | 2.99 | 26.4800 127.6246 |
| $Postp_{it}$ |  |  |  | 51.516 (29.3027) | 1.76 | -6.0433 109.0762 |
| $(POLICY_{it}*Postp_{it})$ |  |  |  | 31.129 (45.799) | 0.68 | -58.8347 121.0932 |
| R-squared | 0.7691 |  |  | 0.7860 |  |  |
| Adj R-squared | 0.7643 |  |  | 0.7643 |  |  |

Note: Standard errors in parentheses

** = p< .01

*** = p< .001.

**Table 4. Moderation analysis.**

|  | Model 1 | Model 2 | Model 3 | Model 4 | Model 5 |
|---|---|---|---|---|---|
| $POLICY$ | 17.528(24.639) | 39.871(25.7659) | 39.871 (25.7659) | 40.942(26.0542) | 20.9519(27.372) |
| $x_{it1}$ | 36.1326(21.814) | - | 82.568(41.2074)* | 18.971(23.7823) | 11.5452 (50.5391) |
| $x_{it2}$ | 135.8672(27.3306)*** | 98.785(28.5927)** | 16.216(46.10326) | 116.907(27.8037)*** | 114.099(53.4181)*** |
| $x_{it3}$ | 107.5747(18.2715)*** | 96.506(37.7114)** | 13.938 (54.7239) | 116.6927 (26.947)*** | 108.219 (55.7793)*** |
| $y_{it}$ |  | 2.459 (0.6565)**** | 2.459 (0.6565)*** | 2.241 (1.0009)* | 2.874 (0.6867)*** |
| $z_{it}$ |  | -0.003 (0.0008)** | -0.003(0.0008)** | -0.003 (0.0008)** | -0.002 (0.0008)** |
| _cons |  | -82.568 (41.2074)* |  |  |  |
| $i.POLICY\#\#^*c.y_{it}$ |  |  |  | 0.317 (1.1018) |  |
| $i.POLICY\#\#^*c.z_{it}$ |  |  |  |  | -0.003(0.0014)* |
|  | $F_{(4, 548)} = 22.15$ | $F_{(5, 546)} = 6.39$ | $F_{(6, 546)} = 18.01$ | $F_{(7, 545)} = 15.42$ | $F_{(7, 545)} = 16.1$ |
| Prob > F | 0.000 | 0.000 | 0.000 | 0.000 | 0.000 |
| R-squared | 0.7411 | 0.5535 | 0.7411 | 0.7680 | 0.7961 |
| Adj R-squared | 0.7316 | 0.5266 | 0.7316 | 0.7571 | 0.7843 |
| Average marginal effects |  |  |  |  |  |
|  |  | at |  | dy/dx Std. Err |  |
|  |  | 1 |  | 82.644 (33.39375)* |  |
| 1.POLICY |  | 2 |  | 20.952 (27.3720) |  |
|  |  | 3 |  | -40.740 (47.7073) |  |

Note: Standard errors in parentheses

*p< .05

**p< .01

***p< .001.

exportations (EXPORT) ($\gamma_2$ = 135.867, t = 4.97, p< .001; $\gamma_3$ = 107.575, p< .001). Model 2 (Table 3) reports estimation results of DIDM.

The coefficients of the time variable (*Postp*), *POLICY*, and the interaction term (*POLICY$_{it}$*\**Postp$_{it}$*) are all statistically insignificant, suggesting that BRI policy implementation has not positively influenced B2C exportations of Chinese Smartphones via CCBECM in BRI countries. This finding is inconsistent with the hypothesis (H1). Therefore, hypothesis (H1) failed to be supported.

## 5. Moderation analysis

To evaluate the moderating effect of our two macroeconomic variables, namely: Internet access rate (IAR ($y_{it}$)) and Gdp per capita (GDP ($z_{it}$)) on exportations (*EXPORT*), we proceeded in hierarchic ways (additively). Table 4 (Model 1) shows the results for the regression that includes only the *POLICY* (Independent variable) and countries development level variable (Status ($\gamma x_{it}$)) (control variable).

In Model 2, we included all direct effects: That is to say, the independent Variable, the control variable, and our two moderators by withdrawing one of the levels e.i, $x_{it1}$ of the categorical variable and running the model with the constant term. However, in Model 3, we have included all of variable of Model 2 and added the level ($x_{it1}$) of the categorical Variable excluded in model 2 and ran the model without the constant term. We did so to demonstrate the equality relationship between the constant term and the level of the categorical variable excluded in model 2. As we can see from Model 2 and Model 3, the coefficient of the ($x_{it1}$) is equal to the constant term ($x_{it1}$ = cons = -82.568 (41.2074), p< .05)

From Model 4, we started to evaluate the interaction between *Policy* (Independent Variable) and our two moderators (i.e., Internet access rate (IAR ($y_{it}$)) and GDP per capita (GDP ($z_{it}$))). In that attempt, our goal is to ascertain Exportations ((Y)) from the impact of *BRI$_{it}$* with IAR ($y_{it}$) and GDP ($z_{it}$) serving as moderators of that relationship. In other words, we hypothesize that the relationship between our independent variable (*POLICY$_{it}$*)and dependent variable Exportation (Y) is moderated by IAR ($y_{it}$) and DGP ($z_{it}$). Our focal independent variable *Policy$_{it}$* is a binary variable, while IAR ($y_{it}$) and DGP ($z_{it}$) are continuous variables. As IAR ($y_{it}$) and DGP ($z_{it}$), used to create the interaction terms with independent variable *POLICY$_{it}$* are all continuous variables and have no meaningful zero points associated with them; the simple slope tests for these predictors may be difficult to understand if zero does not fall within the range of possible values, even though the regression slope associated with the interaction terms remain significant. If zero is not a substantial value on these predictors, the intercept is meaningless during interpretation. To avoid that, we first centered (mean centering) our two moderating variables before running the interaction analysis. Next, we used the centering variables to run the interaction analysis between our *POLICY$_{it}$* and IAR ($y_{it}$) and DGP ($z_{it}$), respectively. Those analyses are reported in Table 4 (models 4 and 5).

In model 4, we see that the interaction term between *POLICY$_{it}$* and IAR ($y_{it}$) was not statistically significant (b = 0.317, s.e. = 1.1018, p>.05), suggesting that the relationship between the influence of BRI policy and exportations (EXPORT) is not conditional on the level of Internet Access Rate (IAR), which is inconsistent with the hypothesis (H2). Accordingly, the hypothesis (H2) failed to be supported. Next, we can see from model 5 that the interaction term between *POLICY$_{it}$*, and DGP ($z_{it}$), was negatively significant (b = -0.003, s.e. = 0.0014, p < .05), indicating that the effect of GDP (GDP per capita) is stronger on exports from Non BRI countries than BRI countries. That means the increase of the BRI policy effect will decrease the significant effect of GDP on exports. In other words, If the effect of GDP is negative on exports, its effect will be less negative with increasing the effect of the BRI policy. However, if the effect of

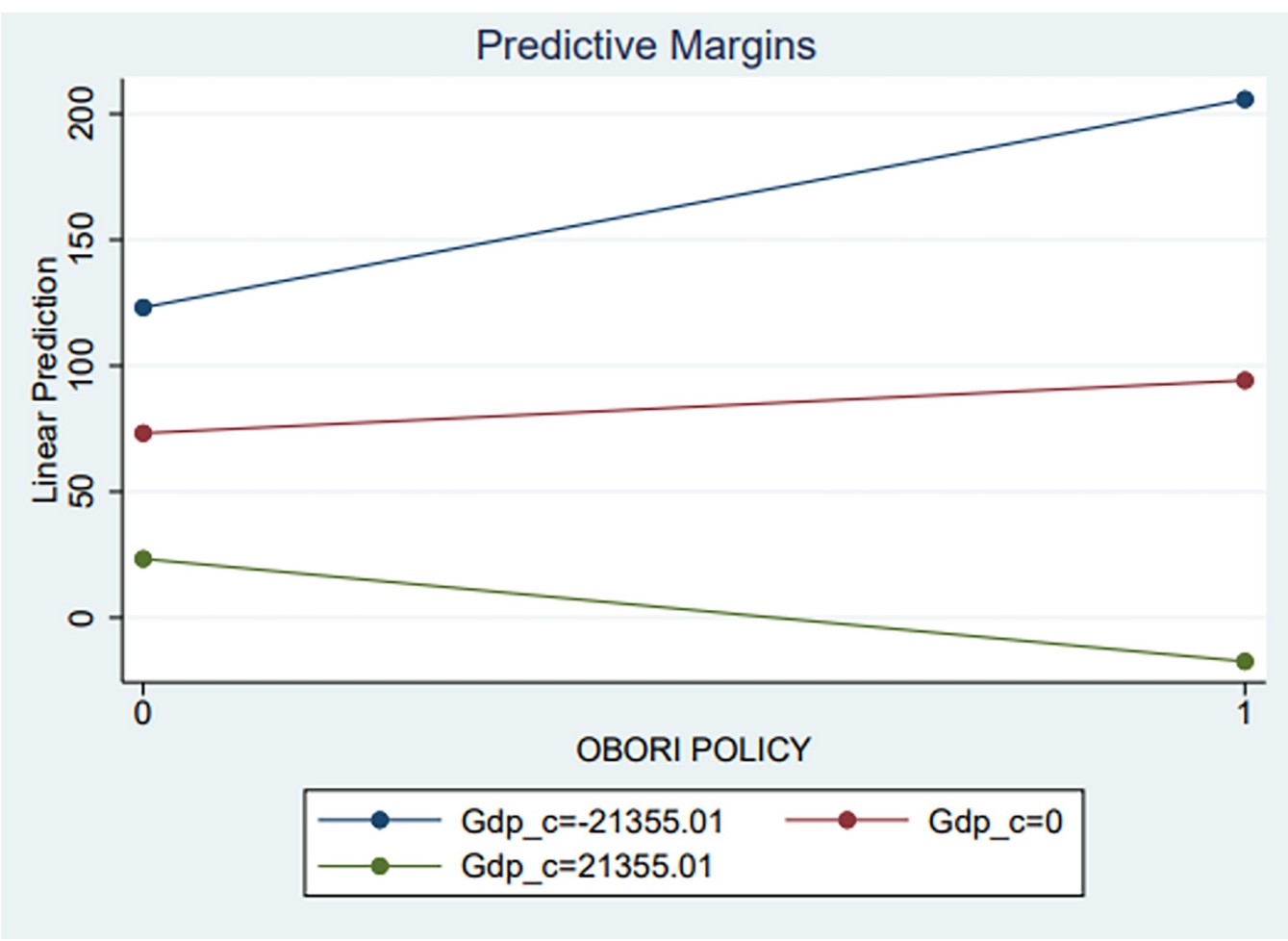

**Fig 3. The interaction term between BRI policy (OBORI Policy) and GDP.**

GDP is positive, the GDP effect will be less positive with increasing the effect of the BRI policy. As a result, this finding is consistent with the hypothesis (H3). Accordingly, hypothesis (H3) is supported. Since this was significant, we probed the interaction to visualize the interaction between BRI policy and GDP at -1sd, at mean, and at +1sd on GDP (Fig 3). The blue line represents the predicted relationship between BRI policy and exports at 1sd below the mean on $GDP\_c$ (mean centring). The green line shows the predicted relationship between BRI policy and exportations at 1sd above the mean on $GDP\_c$. And the red line is the predicted relationship between BRI policy and exportations at the mean on the $GDP\_c$ variable. We can see that at 1sd below the mean, the predicted relationship between BRI policy and exportations on $GDP\_c$ is a fairly pronounced negative relationship. At 1sd above the mean, the predicted relationship between BRI policy and exports on $GDP\_c$ has a fairly pronounced positive relationship. And at the mean, the predicted relationship between BRI policy and exportations on GDP is not much pronounced.

## 6. Discussion

The insignificant effect of BRI on exportations revealed in this study is opposed to the claims of most scholars, such as Hou et al. [2], who stated that implementing the BRI has contributed

to boosting the exports of Chinese smartphones via CCBECM. Accordingly, as one may see from this study, the opinion of Hou et al. [2] is not supported empirically. That opinion is founded on a biased appreciation since previous studies have ignored empirically studying the BRI policy impact on the exportations of Chinese smartphone brands through CCBECM. As shown in Fig 2, the BRI policy seems to impact the exportation of Chinese smartphones to BRI countries. However, the level of those exportations is still lower compared to those from non-BRI countries. Indeed, even if the BRI could be an excellent opportunity for Chinese international trade in the future, its current impact on exports of Chinese smartphone brands is still low, as demonstrated in this study. The reason behind those insignificant outcomes could be explained by the underdevelopment level of most BRI countries. That shows that the BRI policy must include more developed and economically strong countries to influence the exports of Chinese smartphones to BRI countries significantly and offer to these countries the same advantages offered to BRI countries as part of the BRI implementation. In doing so, these developed and economically strong countries could export more Chinese smartphone brands.

Regarding the moderating roles of our two main macroeconomic variables, the result showed a significant and negative moderating role of GDP per capita between BRI policy and exports. However, IAR was found to have an insignificant moderating role between BRI policy and exports. These two different results show that GDP per capita strength explains exportations of Chinese smartphone brands in BRI countries better than IAR. That result somehow highlights the strength of GDP per capita in CBEC transactions. Several previous studies, such as the studies of Nolan et al [49], Li et al. [50], and Gibbs et al [51], have already pointed out that. These previous studies have all pointed out the importance of GDP per capita in international electronic transaction studies because the economic level measured by GDP per capita is fundamental for e-shopping practice in a nation [51]. Thus, by introducing GDP per capita as a moderator between BRI policy and exportations, the study contributes to the BRI literature by showing that if the BRI is adequately implemented with all its support projects, such as the construction of port and socio-economic infrastructures, the effect of BRI could significantly reduce the impact of GDP per capita on Chinese product exports through CCBECM in the BRI countries.

## 7. Theoretical contributions and implications

As far as BRI literature is concerned, the contribution of this study is manifold. First, unlike previous studies, our study is not based on mere opinions but on empirical evidence instead.

Second, our study provides a solid theoretical contribution to the literature since there is limited research on the relationships between BRI policy and exports of Chinese smartphones through CCBECM in BRI countries since BRI policy implementation.

Third, to the best of our knowledge, almost no studies focused on GDP per capita and IAR as moderators in analyzing the impact of the BRI policy on exports of Chinese smartphones. The study contributes by shedding light on the moderating role of GDP per capita and IAR in the relationship between BRI policy and exportations of Chinese smartphone brands. Thus, our theoretical framework allows decision-makers to better understand the underlying factors of Chinese smartphone exportations through CCBECM in BRI countries. Therefore, by introducing GDP per capita as a moderator between BRI policy and exportations, the study contributes to the BRI literature by showing that the higher the BRI effect, the less GDP per capita will influence Chinese smartphone product exportations in BRI countries.

## 8. Conclusion and limitations

Utilizing DIDM to quantify the impact of the BRI on Chinese product brand exportations, the analysis showed that the BRI policy's effect needs to be more significant to contribute to

Chinese smartphone exportations within BRI countries. As shown in Fig 1, Chinese smartphone exportations have increased in BRI countries since BRI implementation. However, that growth across BRI countries is less than that of countries non-members of the BRI policy. Therefore, if the goal of the BRI is to contribute to strengthening the sale of Chinese smartphone brands, It will be necessary to promote that policy toward the developed countries since it is projected that the BRI will influence international commerce. Our model offers a more unobstructed view of the actual effect of the BRI on the exportations of Chinese smartphone brands in BRI countries. The study gives decision-makers a clear idea of the impact of BRI policy and highlights the necessary conditions to improve that impact.

However, despite this study's contribution to the literature, it has certain limitations. The first limitation of this work concerns the source of the data. Indeed, the data utilized comes from a single Chinese international online sales platform. The second limitation of this work concerns the official number of BRI countries utilized in this current study, e.i., 28 countries. New countries continue integrating the BRI policy with the geopolitical and economic dynamics. However, we could not consider those countries in this study since we only considered the BRI countries' list available. The third limitation is the product category utilized in this study (only smartphones) since Chinese companies and brands make and sell various products to international buyers through the CCBECM.

As a result, additional research could be conducted by collecting data from other or several Chinese international sales platforms with several product categories and incorporating the omitted countries in this current research to verify our results. Moreover, the data could affect our analysis and conclusions (2012–2019). First, the COVID-19 pandemic and second, the recent Russian invasion of Ukraine has also produced influences. That means policymakers would also need more recent analyses after the pandemic, from 2020–2023.

Even though several scholars claimed that the implementation of the BRI had contributed to the growth of CCBECM and provided excellent business opportunities for BRI countries in terms of easy access to Chinese products, such as smartphones, this study points out that that growth in terms of smartphone exportations to BRI countries is less than that of countries that are not members of the BRI. Therefore, as BRI nations now account for only 18% of the global GDP, promoting the BRI toward developed countries will be essential if the policy's objective is to strengthen the export of smartphones globally. Therefore, decision-makers are recommended to improve their strategies by expanding the number of BRI nations to increase the global adoption of Chinese smartphones.

## Supporting information

**S1 File.**
(XLSX)

## Author Contributions

**Data curation:** Karamoko N'da.

**Formal analysis:** Karamoko N'da.

**Investigation:** Karamoko N'da.

**Methodology:** Karamoko N'da.

**Software:** Karamoko N'da.

**Supervision:** Jiaoju Ge, Steven Ji-Fan Ren, Jia Wang.

**Validation:** Jiaoju Ge, Steven Ji-Fan Ren.

**Writing – original draft:** Karamoko N'da.

**Writing – review & editing:** Jiaoju Ge.

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
