## [Decision Letter · Decision Letter 0]

6 Mar 2023

PONE-D-22-34618Understanding the One Belt One Road Initiative Policy (BRI) influence on exportations of Chinese smartphone brands in BRI countries: The moderating role of the GDP per capitaPLOS ONE

Dear Dr. N'da,

Thank you for submitting your manuscript to PLOS ONE. After careful consideration, we feel that it has merit but does not fully meet PLOS ONE’s publication criteria as it currently stands. Therefore, we invite you to submit a revised version of the manuscript that addresses the points raised during the review process.

We look forward to receiving your revised manuscript.

Kind regards,

László Vasa, PhD

Academic Editor

PLOS ONE

Journal Requirements:

6. Please amend either the title on the online submission form (via Edit Submission) or the title in the manuscript so that they are identical.

Reviewers' comments:

Reviewer's Responses to Questions

**Comments to the Author**

1. Is the manuscript technically sound, and do the data support the conclusions?

Reviewer #1: Yes

Reviewer #2: Partly

2. Has the statistical analysis been performed appropriately and rigorously? 

Reviewer #1: Yes

Reviewer #2: Yes

3. Have the authors made all data underlying the findings in their manuscript fully available?

Reviewer #1: Yes

Reviewer #2: Yes

4. Is the manuscript presented in an intelligible fashion and written in standard English?

Reviewer #1: Yes

Reviewer #2: No

5. Review Comments to the Author

Reviewer #1: Comments and Suggestions for Authors

First of all, I appreciate the opportunity to review your paper, which title is:

Understanding the One Belt One Road Initiative Policy (BRI) influence on exportations of Chinese smartphone brands in BRI countries: The moderating role of the GDP per capita

The topic is very interesting and important. The problem presented with the results is very useful, however, the paper has some shortcomings:

Title: The title is a little bit too long, maybe the short title is better.

Abstract is well-structured and clear at the present form, but there are some missing parts. Rather, the focus should be on the importance of the topic and the relevant (or main) issue more. The research objectives are clear, but the scientific gaps and purpose are unclear or vague. It should be more focused on the importance of the topic and main issues and scientific gaps.

Introduction is well-structured, clear and well-written at the present form. In addition in the introduction some new literature sources should be cited especially in the following areas.

a) Main scientific issues in the field, scientific uncertainties and gaps between the theories.

b) I suggest incorporating the newest debates of prestigious international journals with foreign authors also especially in the field of OBORI.

The subject is actual, and the study contributes to the development of knowledge. The first impression of reading the introduction is that it is brief, but informative and focusing on the narrowly defined topic.

The literature review. The literature review is moderate and adequately supports why the impact of the OBORI policy on CCBECM should be investigated. This is clear, but the justification of why CCBECM is important for international science still needs to be demonstrated.

The three hypotheses were adequately supported by the references and they are acceptable.

Analysis section:

Please harmonize the following sections as part of Material and Methods.

3. Research Context, Data description and Variable measurements. 4. Data description and Variables.

Please correct or explain the followings:

a) The exact relationships of the explanatory variables and the relationship between the explanatory variables and the control variables.

b) Please explain how it can be verified that the presented and measured changes in the dependent variable were created as a result of the independent variables, that is, as a result of the OBORI policy and not as a result of spontaneous market growth? What is the evidence that the OBORI policy has a positive effect? It is understandable that the support processes have a positive effect, but what is the evidence for this, since it is not possible to compare the measured data with a control group. Maybe there is a trend level difference between the period before and after the OBORI policy? Formulate your answer taking into account the following statement: Utilizing DIDM to quantify the impact of the OBORI on Chinese product brand exportations, the analysis showed that the OBORI policy's effect is not significant enough to contribute to Chinese smartphones' exportations within OBORI countries.

Research findings (Conclusions and limitations) –

Conclusions are very well-structured and clear at the present form. The conclusion section, limitations of the model and future research directions are satisfactory at a high level. Please also make further suggestions for policy makers. How can stochastic factors affect the presented models and which factors can indicate uncertainty behaviour?

The paper has very good potential, but the authors need to make a minor revision at this stage. I’m very glad to have the opportunity to read your work.

Recommendation: Minor Revisions Required

Reviewer #2: The research paper analyses a contemporary and exciting topic of Chinese smartphone export tendencies. The approach and the methods applied are unique, the research itself fills a gap in the field, so the paper can be regarded as original.

The abstract is almost perfectly written, however, I would extend it with one first sentence, which indicates the reason and context, why this study was started. The title is appropriate, reflecting the content.

In the introduction, the context and the research questions, research gaps are highlighted. The hypotheses are developed based on the broad and comprehensive literature review, however, this latter rather focuses on the OBOR and trade issues the than introducing the characteristics of the smartphone industry and its export. I recommend to extend the review towards these topics.

Regarding methodology, first the methods then the datasets should be demonstrated, so a restructuring is needed here.

The results are well demonstrated, conclusions are clears, limitations are set.

The text should be checked by a native proofreader and also the terminology should be harmonized. E.g. the authors used the abbreviation OBORI while internationally it is used OBOR.

6. PLOS authors have the option to publish the peer review history of their article (what does this mean?). If published, this will include your full peer review and any attached files.

Reviewer #1: **Yes: **Dr. Gyenge Balázs

Reviewer #2: No

---

## [Author Response · Author response to Decision Letter 0]

27 Oct 2023

Reviewer #1: Comments and Suggestions for Authors

First of all, I appreciate the opportunity to review your paper, which title is:

Understanding the One Belt One Road Initiative (BRI) influence on exportations of Chinese smartphones: The moderating role of the GDP per capita. The topic is very interesting and important. The problem presented with the results is very useful, however, the paper has some shortcomings:

1.Title: The title is a little bit too long, maybe the short title is better.

Thank you for your suggestion about the title of the article. Considering your suggestion, we have shorten the article title. The title is now: " Understanding the One Belt One Road Initiative (BRI) influence on exportations of Chinese smartphones: The moderating role of the GDP per capita"

2. Abstract is well-structured and clear at the present form, but there are some missing parts. Rather, the focus should be on the importance of the topic and the relevant (or main) issue more. The research objectives are clear, but the scientific gaps and purpose are unclear or vague. It should be more focused on the importance of the topic and main issues and scientific gaps.

We sincerely appreciate your positive comments on well-structured and clarity of the Abstract. Regarding the focused of the Abstract that should be on the importance of the topic and main issues and scientific gaps, we have improved the abstract's quality based on your comments in this revised manuscript as follows: 

The One Belt One Road Initiative (BRI) has been the subject of multitudinous studies from various angles. Most previous studies have focused on BRI's economic, geopolitical, or commercial implications for China. However, the few studies that focused on BRI's influence on the exportations or importations of Chinese products via the Chinese Cross-border Electronic Commerce Market (CCBECM) have been carried out based only on authors' opinions rather than on empirical evidence. Therefore, the actual effect of BRI on the exportations of Chinese product brands via CCBECM in BRI countries still needs to be discovered. 

Utilizing B2C exportation data of Chinese smartphone brands and a Difference-in Difference Model (DIDM), we have first examined the actual and direct impact of BRI policy on Chinese smartphone brands exportations via the Chinese Cross-border Electronic Commerce Market (CCBECM) from 2012 to 2019 in BRI countries. Secondly, we assessed the moderating role of GDP per capita (GDP) and Internet Access Rate (IAR) between BRI policy and exportations of Chinese smartphone brands. The results showed that the impact of BRI remains insignificant on the exportations of Chinese smartphones via CCBECM in BRI countries. However, it could be significant if BRI includes more developed and economically strong countries. The study also highlighted a negative moderating role of GDP per capita between BRI policy and exportations, showing that the higher the BRI effect is, the less GDP per capita will influence Chinese smartphone exportations in BRI countries.

3. Introduction is well-structured, clear and well-written at the present form. 

Thank you for your kind comments

In addition in the introduction some new literature sources should be cited especially in the following areas.

a) Main scientific issues in the field, scientific uncertainties and gaps between the theories.

Thank you for your constructive suggestions. First of all, regarding the main scientific issues in the field of BRI, we have already highlighted in the introduction that the advent of BRI has triggered numerous implications in several fields, so that studies on the influence of BRI have been addressed from several scientific angles. To that end we wrote: ("Initially announced by President Xi Jinping as a financial and trade cooperation project between China and countries members [3-5], BRI policy has been studied by many researchers from various angles, such as the geopolitics angle [6-7-8-9-10-11], economic and financial angle [12-13-14-15-16-17], infrastructural angle [18-19], cultural angle [20-21-22], energetic angle [23-24], environmental angle [25-26], and domestic politics angle [27].") 

Regarding the scientific uncertainties on the BRI, we have already pointed out that the only main scientific uncertainty about the BRI in the literature is the exact number of BRI members, which is unknown until now. Moreover, we have highlighted this out in Section 3.2 Data Description and Variables. Thank you once again for these suggestions. 

Finally, regarding the gaps between the theories and visions of the BRI policy, our introduction already pointed out that two prominent opinions (Studies or theories) confront each other: 

The first group of studies claim to see hidden diplomatic-geopolitical ambitions behind the BRI project [7-8]. Those studies claimed that the BRI policy stake is to assert a more active foreign policy based on the new vision of the Chinese world order, which is interconnected capitalism. Therefore, BRI could be seen as an effort to strengthen China's diplomatic and political presence in BRI countries [28]. It should be noted that the supporters of this vision are foreign scholars (non-Chinese citizens, such as [7 and 8]). 

However, the second group of studies pointed out that the BRI primarily aims to support China's domestic economic growth by finding new outlets and opportunities for Chinese industry and product brands [2-5]

b) I suggest incorporating the newest debates of prestigious international journals with foreign authors also especially in the field of OBORI.

Thank you for your constructive suggestion. We really appreciated this suggestion. Therefore, we have done our best to introduce in the introduction, as you suggested, the debate on the alternative to the BRI proposed by the G7 in Germany on June 26, 2022, during the penultimate G7 summit as follows: "…… This point of view is also that of several world organizations led by the West, such as the G7, which describes the BRI as one of the essential links in "debt trap diplomacy" [29]. Therefore, on June 26th, 2022, during the penultimate G7 summit, the group of seven most industrialized countries pledged to raise about US$600 billion in funds over the next five years to counter the BRI initiative and replace it with the newly global project named "Partnership for Global Infrastructure and Investment" to finance needed infrastructure in developing countries [29]". It should be noted that very few prestigious international journals with foreign authors have written on that matter.

The subject is actual, and the study contributes to the development of knowledge. The first impression of reading the introduction is that it is brief, but informative and focusing on the narrowly defined topic.

Thanks for your helpful comments in revising the manuscript

4. The literature review. 

a1/ The literature review is moderate and adequately supports why the impact of the OBORI policy on CCBECM should be investigated. This is clear, but the justification of why CCBECM is important for international science still needs to be demonstrated.

Thanks for your constructive comments in revising the manuscript. Based on your comments we justified why CCBECM is important for international science and trade as follows: 

As an international commercial trade between companies or individuals from various nations using international electronic marketplaces to purchase and sell while relying on the global logistics network to deliver goods, CBEC results in significant cost savings on transaction costs [33]. Its substantial savings on transaction costs are most often related to communications, market research, and administration. Encouraged in emerging economies like China, CBEC has since enabled Chinese exporters and brands to overcome the constraints linked to isolation from potential markets, limited access to information, and high entry costs in the traditional international market [33]. With this, the CCBECM generated about 11 trillion RMB from its advent to 2019, thus creating thousands of dozens of jobs in China and outside China (as most of the sellers on CCBECMs, like Alibaba, AliExpress, Etc., are sellers from outside China). As a result, CCBECM has become a vital transaction means worldwide. Its impact on international trade was even more significant during the COVID-19 outbreak, when CCBECM had become almost the only international transaction framework in China, allowing suppliers to meet buyers' needs worldwide. It is, therefore, an industry essential to global and Chinese economic growth.

a2/ The three hypotheses were adequately supported by the references and they are acceptable.

Thank you for your kind comments in revising the manuscript.

5. Analysis section

Please harmonize the following sections as part of Material and Methods.3. Research Context, Data description and Variable measurements. 4. Data description and Variables.

Thank you for your kind suggestion in revising the manuscript. Based on your comments, we have harmonized the Analysis section. We now have: 3. Method and Material; 3.1 Research Methodology and Theoretical Model; 3.2 Research Context, Data description and Variable measurements; 3.3 Data description and Variables…… 

6. Please correct or explain the following:

a1) The exact relationships of the explanatory variables 

As highlighted in the manuscript, we have 3 explanatory variables. That is: 

1. The explanatory variable (Policy): It measures whether a country is a member of the BRI policy or not. Among countries covered by the data set, 28 BRI countries represent the treatment group (coded = 1), and 41 non-BRI nations represent the control group (coded = 0). 

2. The year (2012–2019): It includes pre- and post-BRI periods. The year 2014 is regarded as the starting year of the BRI. We coded the time: time = 0 if the year is a pre-BRI period; time = 1 if the year if t is a post-policy period. 

3. The interaction term between the groups and time. The relationship between these 3 explanatory variables is obvious since policy implementation took place over several years (2014–2019 in the case of this study). Therefore, taking into account the periods before and after the implementation of the policy, as well as the interaction term between the groups and time, is necessary to judge the relevance or effect of this policy on international transactions.

a2) The relationship between the explanatory variables and the control variables.

As Control variables, the study incorporated 3 variables: That is 

1. Country development level ("Status"). 

2. Internet Access Rate (IAR), 

3. GDP per capita (GDP) 

The country's development level was measured by the World Bank classification of the country's development levels. Thus, Lower Middle-Income countries and lower-income countries are coded as Developing countries and equal to 1; Middle-Income countries are coded as Emerging countries and equal to 2; and higher-income countries are coded as Developed countries and equal to 3. 

Regarding the relationship between the explanatory variables and the control variables, although we have not run a correlation matrix to show the relationships between explanatory variables and the control variables of the study, one may see from the mode1 and model 2 (Table 3) the changes in the value of coefficients of "Policy" variable and the control variable of countries development level "status". Moreover, models 4 and 5 (Table 5) give us an idea of the relationships between the study's explanatory and control variables.

b) Please explain how it can be verified that the presented and measured changes in the dependent variable were created as a result of the independent variables, that is, as a result of the OBORI policy and not as a result of spontaneous market growth. 

Thank you for this question. First, we believe this question has already been answered in Section 3.3, Research Methodology and Theoretical Model. We have explained in that section that the implementation of the BRI did not provide a significant impact on the export trends of smartphones since the second assumption of the DID model showed that the exportations of BRI and non-BRI countries (NBRI) have experienced almost the same exportation trends before and after implementing the BRI. One may observe that after the implementation of BRI (2014–2019), the exportations of the treatment group (BRI countries) and control group (non-BRI countries) have followed almost the same trends. However, exportation in non-BRI countries (control group) was higher than in the treatment group (BRI countries). That is why we further ascertained the possible source of the differential influence of BRI policy on exports between BRI countries and non-BRI countries based on countries' socio-economic indicators after running the DID model. To this end, we evaluated the moderating effect of two main socio-economic variables (IAR and their GDP per capita) on exportations. 

7. What is the evidence that the BRI policy has a positive effect? 

Thank you for this question. First of all, this study did not conclude that BRI policy positively affects the exportation of Chinese smartphones towards BRI countries. We clearly stated that Utilizing DIDM to quantify the impact of the BRI on Chinese smartphones' exportations, the analysis showed that the BRI policy's effect is not significant enough to contribute to Chinese smartphones' exportations within BRI countries. In other words, an insignificant effect of the BRI policy on Chinese smartphone exportations within BRI countries has been found. 

8. Research findings (Conclusions and limitations) 

Conclusions are very well-structured and clear at the present form. The conclusion section, limitations of the model and future research directions are satisfactory at a high level. 

Thank you for your kind comments in revising the manuscript.

Please also make further suggestions for policy makers. 

Even though several scholars claimed that the implementation of the BRI had contributed to the growth of CCBECM and provided excellent business opportunities for BRI countries in terms of easy access to Chinese products, such as smartphones, this study points out that that growth in terms of smartphone exportations to BRI countries is less than that of countries that are not members of the BRI. Therefore, as BRI nations now account for only 18% of the global GDP, promoting the BRI toward developed countries will be essential if the policy's objective is to strengthen the export of smartphones globally. Therefore, decision-makers are recommended to improve their strategies by expanding the number of BRI nations to increase the global adoption of Chinese smartphones”.

9. How can stochastic factors affect the presented models and which factors can indicate uncertainty behaviour?

Thank You for this question. As we have already stated, some stochastic factors, such as the COVID-19 pandemic and the recent Russia-Ukraine war, can influence the presented models and produce influences. Indeed, these factors influenced the geopolitical and international transactions context. That means that the geopolitical context and international transactions have generated uncertain behaviour among potential exporters or buyers of Chinese smartphones through CCBECMs. That is why we have pointed out that policymakers would need more recent analyses after the pandemic and the Russia-Ukraine war (2020–2023).

The paper has very good potential, but the authors need to make a minor revision at this stage. I’m very glad to have the opportunity to read your work.

Thank you for your kind comments and suggestions in revising our manuscript. We are also happy to have you review our manuscript. Thank you once again for all your suggestions and comments, which will assist us in enriching and making our manuscript better.

1. Reviewer #2: The research paper analyses a contemporary and exciting topic of Chinese smartphone export tendencies. The approach and the methods applied are unique, the research itself fills a gap in the field, so the paper can be regarded as original.

Thank you for your kind comments in revising the manuscript.

2. The abstract is almost perfectly written, however, I would extend it with one first sentence, which indicates the reason and context, why this study was started. 

We sincerely appreciate your positive comments on the perfect writing of the abstract. Based on your suggestions, we have improved the abstract's quality by indicating the reason and context of why this study was started in this revised manuscript as follows: 

The One Belt One Road Initiative (BRI) has been the subject of multitudinous studies from various angles. Most previous studies have focused on BRI's economic, geopolitical, or commercial implications for China. However, the few studies that focused on BRI's influence on the exportations or importations of Chinese products via the Chinese Cross-border Electronic Commerce Market (CCBECM) have been carried out based only on authors' opinions rather than on empirical evidence. Therefore, the actual effect of BRI on the exportations of Chinese product brands via CCBECM in BRI countries still needs to be discovered. 

Utilizing B2C exportation data of Chinese smartphone brands and a Difference-in Difference Model (DIDM), we have first examined the actual and direct impact of BRI policy on Chinese smartphone exportations via the Chinese Cross-border Electronic Commerce Market (CCBECM) from 2012 to 2019 in BRI countries. Secondly, we assessed the moderating role of GDP per capita (GDP) and Internet Access Rate (IAR) between BRI policy and exportations of Chinese smartphone brands. The results showed that the impact of BRI remains insignificant on the exportations of Chinese smartphones via CCBECM in BRI countries. However, it could be significant if BRI includes more developed and economically strong countries. The study also highlighted a negative moderating role of GDP per capita between BRI policy and exportations, showing that the higher the BRI effect is, the less GDP per capita will influence Chinese smartphone exportations in BRI countries. 

3. The title is appropriate, reflecting the content.

Thank you for your kind comments.

4. In the introduction, the context and the research questions, research gaps are highlighted. The hypotheses are developed based on the broad and comprehensive literature review, however, this latter rather focuses on the OBOR and trade issues than introducing the characteristics of the smartphone industry and its export. I recommend to extend the review towards these topics.

Thank you very much for these constructive suggestions and comments. Based on your recommendation, we extended the review towards the characteristics of the smartphone industry and its exports through CCBECM as follows: In the literature on the relationship between BRI policy and smartphone exports to BRI countries through CBECM, one notes several claims from previous studies without any empirical evidence. For instance, Yang and Asbury [45] claimed the advent of BRI provided great opportunities to Chinese smartphones regarding their purchases and promotions within-country members. In studying different countries' preferences for Chinese smartphone exportations via CCBECM, [2] argued that the advent of BRI has provided great opportunities for purchasing Chinese smartphone brands through CCBECM. However, despite all these claims, we note in the previous literature a lack of empirical studies demonstrating the impact of the BRI policy on Chinese smartphone exportations through the CCBECM in BRI countries.

5. Regarding methodology, first the methods then the datasets should be demonstrated, so a restructuring is needed here.

Thank you for your constructive comments and suggestions in revising the manuscript. Based on your comments and suggestions, we restructured the manuscript. We placed the methods first before the dataset description. 

6. The results are well demonstrated, conclusions are clears, and limitations are set.

Thank you for your kind comments.

4. The text should be checked by a native proofreader and also the terminology should be harmonized. E.g. the authors used the abbreviation OBORI while internationally it is used OBOR.

Thank you for your constructive comments and suggestions in revising the manuscript. We have done our best to correct grammar errors through a proofreader and harmonized the abbreviation OBORI using BRI instead of OBORI, as you suggested. It should be noted that internationally, sometimes, BRI is used instead of OBOR.

---

## [Editor Report · Decision Letter 1]

21 Nov 2023

Understanding the One Belt One Road Initiative (BRI) influence on exportations of Chinese smartphones: The moderating role of the GDP per capita

PONE-D-22-34618R1

Dear Dr. Karamoko N'da,

We’re pleased to inform you that your manuscript has been judged scientifically suitable for publication and will be formally accepted for publication once it meets all outstanding technical requirements.

Kind regards,

László Vasa, PhD

Academic Editor

PLOS ONE
---

## [Editor Report · Acceptance letter]

29 Nov 2023

PONE-D-22-34618R1 

Understanding the One Belt One Road Initiative (BRI) influence on exportations of Chinese smartphones: The moderating role of the GDP per capita 

Dear Dr. N'da:

I'm pleased to inform you that your manuscript has been deemed suitable for publication in PLOS ONE. Congratulations! Your manuscript is now with our production department. 

Kind regards, 

on behalf of

Prof. Dr. László Vasa 

Academic Editor

PLOS ONE